# Elucidation of Antimicrobial Activity of Non-Covalently Dispersed Carbon Nanotubes

**DOI:** 10.3390/ma13071676

**Published:** 2020-04-03

**Authors:** Mansab Ali Saleemi, Mohammad Hosseini Fouladi, Phelim Voon Chen Yong, Eng Hwa Wong

**Affiliations:** 1School of Biosciences, Taylor’s University Lakeside Campus, Subang Jaya 47500, Selangor, Malaysia; mansabalisaleemi@sd.taylors.edu.my (M.A.S.); Phelimvoonchen.yong@taylors.edu.my (P.V.C.Y.); 2School of Engineering, Taylor’s University Lakeside Campus, Subang Jaya 47500, Selangor, Malaysia; mfoolady@gmail.com; 3School of Medicine, Taylor’s University Lakeside Campus, Subang Jaya 47500, Selangor, Malaysia

**Keywords:** carbon nanotubes, sodium dodecylbenzene sulfonate, pathogens, antimicrobial activity

## Abstract

Microorganisms have begun to develop resistance because of inappropriate and extensive use of antibiotics in the hospital setting. Therefore, it seems to be necessary to find a way to tackle these pathogens by developing new and effective antimicrobial agents. Carbon nanotubes (CNTs) have attracted growing attention because of their remarkable mechanical strength, electrical properties, and chemical and thermal stability for their potential applications in the field of biomedical as therapeutic and diagnostic nanotools. However, the impact of carbon nanotubes on microbial growth has not been fully investigated. The primary purpose of this research study is to investigate the antimicrobial activity of CNTs, particularly double-walled and multi-walled nanotubes on representative pathogenic strains such as Gram-positive bacteria *Staphylococcus aureus*, Gram-negative bacteria *Pseudomonas aeruginosa*, *Klebsiella pneumoniae*, and fungal strain *Candida albicans*. The dispersion ability of CNT types (double-walled and multi-walled) treated with a surfactant such as sodium dodecyl-benzenesulfonate (SDBS) and their impact on the microbial growth inhibition were also examined. A stock concentration 0.2 mg/mL of both double-walled and multi-walled CNTs was prepared homogenized by dispersing in surfactant solution by using probe sonication. UV-vis absorbance, Fourier transform infrared spectroscopy (FTIR), and transmission electron microscopy (TEM) were used for the characterization of CNTs dispersed in the surfactant solution to study the interaction between molecules of surfactant and CNTs. Later, scanning electron microscopy (SEM) was used to investigate how CNTs interact with the microbial cells. The antimicrobial activity was determined by analyzing optical density growth curves and viable cell count. This study revealed that microbial growth inhibited by non-covalently dispersed CNTs was both depend on the concentration and treatment time. In conclusion, the binding of surfactant molecules to the surface of CNTs increases its ability to disperse in aqueous solution. Non-covalent method of CNTs dispersion preserved their structure and increased microbial growth inhibition as a result. Multi-walled CNTs exhibited higher antimicrobial activity compared to double-walled CNTs against selected pathogens.

## 1. Introduction

Carbon nanotubes (CNTs), first discovered in 1991 by a Japanese scientist Sumio Iijima [1], are currently considered to be a top-class topic in academic research institutions and several industrial areas. Their impressive physicochemical properties are due to their incredible thermal and electrical conductivity, strong mechanical strength, and high aspect ratio of nanotubes [2,3,4,5,6,7,8]. The antimicrobial activity of carbon nanotubes is strongly based on surface chemistry, which controls some critical factors such as oxidation power or hydrophobicity [9,10]. This part deals with the intrinsic antimicrobial characteristics of CNTs. Here, we comprise features that induce from standard CNTs pretreatment methods; for instance, during purification concentrated oxygenating acids are sometimes used and, at the same time, affect the surface morphology and chemistry of the respective materials. Intentional covalent and non-covalent functionalization of materials during synthesis, which can be applied to alter the antimicrobial features of CNTs. The aqueous dispersion of nanomaterials is another serious issue, as it causes direct impacts on cell surface contact and bioavailability for the pathogen [11]. The pretreatment of the carbon nanomaterials is a crucial step to disperse in an aqueous solution and the introduction of additives that might be useful to stabilize the hydrophobic nature of CNTs in biological medium or an aqueous dispersion [12].

The high aspect ratio of CNTs makes them vulnerable to form a bundle and entanglement. The reason behind CNTs is formed bundle due to a strong van der Waals interaction between nanotubes [7,13]. Such kind of interaction makes the dispersion of CNTs a challenging task by the researchers. Thus, it is necessary to modify/functionalize the carbon nanotubes to enhance their dispersion with the attachment of a functional group by the covalent or non-covalent method. The non-covalent method of functionalization outweighs the covalent method of dispersion because of graphene sheets π-π system (mean external surface area of tubes) remained intact, and the structural properties of CNTs are un-affected [14]. Certainly, the non-covalent method of CNTs dispersion by a surfactant solution is more suitable to use for enhancing dispersions of CNTs because of its simple process of modification, including only probe sonication, centrifugation, and filtration for the sake of preserving CNT’s properties and structures [11,12]. The hydrophobic chain group of the applied surfactants can interact with the CNTs’ sidewalls by hydrophobic interactions and thus anchor the molecules of surfactants to the carbon nanotubes, remaining hydrophilic head interacts with the aqueous phase. The molecules of surfactants are strongly adsorbed on the CNTs surface and inhibit their re-agglomeration so that dispersions of nanotubes could maintain colloidal stability for several months [15,16]. Earlier studies have shown that CNTs are more effective in inhibiting the growth of pathogens (such as *V. parahaemolyticus* and *Escherichia coli*) in their dispersed form than CNT bundles [17,18]. The reason for the high effectiveness of individual nanotubes in inhibiting the microbial growth after dispersion could be because of the increased surface contact chances with the microbial cells (*E. coli, S. typhimurium* and *P. denitrificants*) [19]. The proposed antimicrobial activity of CNTs depends on their state of dispersion.

This work focuses on the modification of CNTs via a non-covalent functionalization method and determines the antimicrobial property of CNTs and also to study the antimicrobial mechanism of action of CNTs on different selected pathogens, such as *Staphylococcus aureus*, *Pseudomonas aeruginosa*, *Klebsiella pneumoniae*, and fungal strain *Candida albicans*. According to our understanding, the antimicrobial activity of DWCNTs to *Klebsiella pneumoniae* has not been investigated. We have been observed that both CNT types (double-walled and multi-walled) have broad-spectrum antimicrobial effects. Also, this study has provided such a single platform to discuss the biocompatibility of surfactant (SDBS) and the antimicrobial activity of functionalized-CNTs to different selected pathogens. Previously, different ranges of surfactants solution have been studied for the CNTs’ dispersion, such as sodium dodecyl sulfate (SDS) [16], octyl phenol ethoxylate (Triton X-100) [20], hexadecyltrimethylam-monium bromide (CTAB) [7,21], sodium dodecyl-benzenesulphonate (SDBS) [13], etc. The surfactant (SDBS) was applied in this study to modify carbon nanotubes by attaching a functional group on the surface and improves its aqueous phase dispersion. Besides that, the antimicrobial activity of CNTs (DWCNT and MWCNT) treated with surfactant were also reported against selected pathogens. CNT’s antimicrobial activity was observed after analyzing the OD growth curves and viable cell count.

## 2. Materials and Methods

### 2.1. Collection and Purification of CNT Samples

DWCNTs and MWCNTs were obtained from (NE Scientific Enterprise, Kuala Lumpur, Malaysia) with a median outer diameter of 2–4 nm, length 10–20 µm, and purity 90% for DWCNTs and a median outer diameter of 10–15 nm, length ~100 µm, and purity > 95% of MWCNTs. They synthesized the samples using the chemical vapor deposition method. The method used for the purification of CNTs is as follows. 100 mg of raw CNTs (DWCNTs and MWCNTs) were heated at 450 °C for 90 min at room temperature. After heating, CNTs were inserted into a conical flask comprising 6 M HCl for the eradication of metallic catalyst (Ni, Fe, etc.). Later, a membrane filter was used for the filtration of the acidic solution, and the filtered nanomaterials were shifted into a conical flask containing 3 M NaOH and further heated at temperature (100 °C) under reflux to eliminate the aluminium oxides [11]. Again, the suspension was passed through the membrane filter, and then distilled water used to wash the filtered nanomaterials until pH becomes neutral. Finally, CNT samples were kept in an oven for drying at 55 °C.

### 2.2. Preparation of Surfactant-Modified CNTs

10 mg of CNTs after purification were suspended with 0.05 wt.% of sodium dodecylbenzene sulfonate (SDBS, purchased from Sigma-Aldrich, St. Louis, Missouri, USA) solution. The chemical structure of SDBS has been shown in (Figure 1). The CNT dispersion was ultrasonicated for 30 min to acquire the SDBS-adsorbed CNT surface, as seen in Figure 2. The resulting suspension was centrifuged for 1 h at 10,000 rpm. The upper supernatant fluid was collected for the characterization purpose with UV-vis spectroscopy. Otherwise, a membrane filter (0.45 μm) was used to filter the solution and then distilled water used to wash the filtered nanomaterials until pH becomes neutral, and the suspension was dried in an oven at 55 °C [13].

### 2.3. Characterization of Surfactant-Modified CNTs

The technique UV-vis spectroscopy was applied to determine the absorbance capacity of CNTs in a surfactant solution, operated at 600 nm. During the initial stage, a pure 0.05% SDBS solution was used for the baseline correction to subtract their absorbance from CNTs’ dispersions. In the second stage, the SDBS-treated CNT was investigated with a corresponding concentration of SDBS. Transmission electron microscopy (TEM, Hitachi Limited, Tokyo, Japan) was used to observe the dispersion of CNTs after treated with the surfactant solution.

For this purpose, prepared suspension of CNTs was re-dispersed in distilled water (DW) at 0.5 mg/mL concentration, and a drop (10 μL) from the mentioned concentration was placed on a transmission electron microscopy (TEM) grid coated with carbon [13]. Then the image can be seen under TEM.

### 2.4. Preparation of Bacterial Cultures

In this experimental study, microbial strains such as Gram-positive bacteria *Staphylococcus aureus* ATCC 25923, Gram-negative bacteria *Pseudomonas aeruginosa* ATCC 15692, *Klebsiella pneumoniae* ATCC 43816, and fungal strain *Candida albicans* ATCC 10231 were used to observe the inhibitory effects of CNTs. The Luria-Bertani (LB, Oxoid) broth was used for the growth of bacterial strains at 37 °C in a shaking incubator with constant agitation at 220 rpm for 15–16 h. Yeast cells were grown on yeast peptone dextrose (YPD, Oxoid) at 28 °C under constant agitation at 220 rpm for 30 h. The microbial culture was passed through the centrifuge machine at 6000 g for 2 min [22]. The cells were washed three times with NaCl (0.9%) for the eradication of constituents from the growth medium and residual macromolecules. The cells were resuspended with 0.9% NaCl for further use.

### 2.5. Evaluation of Surfactant Biocompatibility

Here, 100 mL of bacterial and fungal cultures were used with different concentrations of surfactant solution dissolved in LB and YPD medium in order to determine the biocompatibility of surfactant (SDBS). After treatments, microbial growth was observed by determining the optical density growth curve at 600 nm after every 1 h interval for bacterial strains and 2 h intervals for fungal strain. The interaction between surfactant and microbial cells was examined by treating microbial cells with varying concentrations of SDBS in the medium of YPD and LB. All the CNT antimicrobial experimental studies were carried out at pH > 7.

### 2.6. Treatment of Bacterial Cells with CNTs

Ten-fold serial dilutions (1:10) of cell suspensions were made in 0.9% NaCl to achieve the microbial cell suspension at concentrations of ~10^7^ to 10^8^ CFU/mL. 150 μL of microbial suspensions were added into the centrifuge tubes. The CNTs (20 μL), such as DWCNT and MWCNT with desired concentration (20, 40, 60, 80, 100 μg/mL) were introduced into the Eppendorf tubes. The cell suspensions (150 μL) as a control sample were added into the DI water (20 μL). The Eppendorf tubes were kept spinning on a mixer at 170 runs per minute (rpm) for 1 h.

### 2.7. Measurements of Optical Density (OD)

After treatment for 1 h, the mixtures were taken into the tubes and kept into the 5 mL of YPD and LB medium. In a shaking incubator, the bacterial strains were incubated at 37 °C with continuous agitation at 220 rpm. Yeast cells were incubated at 28 °C under the same constant agitation at 220 rpm. The spectrophotometer was used to measure the optical density at 600 nm after passing every 60 min. As a function of growth time, optical density growth curves were achieved by plotting OD values. The initial viable cell number is directly related to the time corresponding to exponential growth in a sample, which is called growth time. The longer growth time requires to enter the exponential growth phase if the initial viable number of cells is lower in the samples. Hence, the exponential time appearance of growth could be employed as an indicator of the initial viable number of cells in a sample, thus elucidating the functionalized CNTs’ antimicrobial properties to the pathogens.

### 2.8. Determination of Viable Cell Number

First, the cells were treated with various concentrations of functionalized CNTs (DWCNTs and MWCNTs), and a reduction in the number of viable cells was assessed through the conventional method of surface plating. The CNTs-cell and control specimens were diluted (1:10) with 0.9% NaCl solution. The viable cell number from each sample was evaluated by the surface plating of appropriate dilution (0.1 mL) onto the agar plates, such as cetrimide agar plates for *Pseudomonas aeruginosa*, brain heart infusion (BHI) agar plates for *Staphylococcus aureus*, MacConkey agar and yeast peptone dextrose (YPD) plates for *Klebsiella pneumoniae,* and *Candida albican.* After 24 h of bacterial and 48 h of fungal incubation, colonies were counted at 37 °C for bacterial strains and 28 °C for fungal strain; thus, reduction in the number of viable cells was determined as colony-forming units per milliliter (CFU/mL) [22].

### 2.9. SEM Imaging

The biological samples were prepared, and determined the structural changes of microbial cells, treated or not with CNTs (DWCNT and MWCNT) using scanning electron microscopy (SEM). The CNT-cell specimens were passed through a 0.2 microns pore-sized membrane filter (Millipore), quickly fixed with 2.5% of glutaraldehyde for one hour at 25 °C and then post-fixed with osmium tetroxide (1%) after three washes in PBS for one hour at 4 °C. For dehydration, biological samples were passed through the graded series of ethanol (30%, 50%, 70%, 80%, 90%, 95%, and 100% v/v) and dried the samples at 25 °C [23]. After drying, the samples were sputter-coated with gold and observed the morphological changes of microbial cells treated or not with CNTs (DWCNTs and MWCNTs) by field emission scanning electron microscopy (FE-SEM, Hitachi Limited, Tokyo, Japan).

### 2.10. Statistical Analysis

All experimental studies were conducted in triplicate. These values are expressed as the mean ± standard deviation (SD). Single-factor analysis of variance (ANOVA) was applied to evaluate the statistical significance of results, and *p*-value < 0.05 was considered significant.

## 3. Results and Discussion

Individually dispersed carbon nanotubes are active in the UV-vis spectral region. CNTs aggregates do not absorb in this region [16]. Thus, CNTs dispersion can be characterized by using UV-vis absorbance spectroscopy. Figure 3 indicates that purified CNTs were mixed in distilled water by sonication (A, C) and SDBS-treated CNTs (B, D) for 30 min. Likewise, Figure 4A shows the dispersion ability of SDBS-treated CNTs and p-CNTs analyzed using a UV-vis spectrophotometer at 600 nm. The absorbance values were taken at 600 nm based on the previously reported studies [24,25,26,27,28].

Figure 4B indicates the Fourier transform infrared spectroscopy (FTIR) spectra of pure CNTs and SDBS modified-CNTs. Noticeably, the FTIR spectra of SDBS-modified CNTs sample elucidates clear signs of functionalities as compared to pure CNT. However, it has been shown that the spectra of DWCNTs treated with SDBS, the peak was appeared at 2945 and 851 cm^−1^, while the peak of MWCNTs appeared at 2998 and 887 cm^−1^. The DWCNTs peaks at 1129 cm^−1^, whereas MWCNTs peaks at 1171 cm^−1^ are allotted to ionic sulfonate SO^3^−group. The occurrence of all these peaks shows the CNTs’ functionalization by SDBS. Also, the peaks appeared at 3598 and 1725 cm^−1^ could be ascribed to the trace water stretching vibration in CNTs. Besides, FTIR spectra demonstrate different peaks at 1242, 1319, 1382, and 1409 cm^−1^ originated from pure CNTs [29]. This spectrum confirms the recognition of SDBS grafted on the surface of CNTs.

It has been shown that solubility of both types of CNTs (DWCNTs and MWCNTs) was higher after applying a surfactant solution than that in their pure form. The surfactant molecules adsorbed more strongly on the surface of CNTs, which enabled them to suspend in water. This study showed that the chemical structure of surfactant played a very important role in the dispersion of CNTs. For the dispersion of nanotubes in water, the molecules of surfactant orient themselves in such a way that hydrophobic tail groups move to the surface of nanotube, whereas hydrophilic head groups move to the aqueous phase, causing a lowering interfacial tension of the nanotube/water [30]. Therefore, the surfactants’ dispersing power depends on how strongly it adsorbs onto the surface of CNTs. This surfactant (SDBS) consists of a benzene ring structure that adsorbs more firmly to the surface of graphite because of the pi–pi stacking type of interaction [31,32]. In general, the hydrophobic tail groups tend to stick on the surface of graphite because of the methylene units of hydrocarbon chains match well with the graphitic unit cells [33]. Hence, the efficacy of adsorption and subsequently, surfactants’ dispersing power are significantly affected by the surfactants’ tail length. The longer tail shows more steric hindrance and high spatial volume, therefore providing great repulsive forces among different individual nanotubes [34]. Besides that, the surfactant could enhance the dispersion of nanotubes containing unsaturated bonds on their tail groups [34].

### 3.1. Biocompatibility of Surfactant

It has been demonstrated that the applied surfactant (SDBS) provides a high level of CNTs dispersion [20]. This surfactant was not involved in the antimicrobial activity of CNTs at lower concentrations after investigation of its biocompatibility. The antimicrobial activity of surfactant was studied by incubating selected pathogenic strains with the following surfactant concentrations: 0.05 wt.%, 0.5 wt.%, and 1 wt.% followed by the investigation of optical density (OD) growth, as shown in (Figure 5). It has been shown that the antimicrobial property of surfactant (SDBS) depends on concentration and treatment time [11]. It has been observed that a significant antimicrobial activity of surfactant at 0.5 and 1 wt.% after 6 h of treatment with bacterial isolates and after 10 h treatment with fungal strain. However, microbial treatments with the anionic surfactant with 0.05 wt.% have been demonstrated to have no significant influence on the cell viability and the time required to reach the exponential phase growth was almost similar as compared to the control sample. Furthermore, the OD growth curves of incubated *Staphylococcus aureus* (Figure 5A) and *Pseudomonas aeruginosa* (Figure 5B) with 0.05 wt.% of SDBS exhibited minor toxicity effect, while *Klebsiella pneumoniae* (Figure 5C), and *Candida albican* (Figure 5D) results indicated that no significant toxic effect with the same concentration.

It can be concluded that negatively charged surfactant molecules do not interact with the negatively charged lipid membrane of the pathogens, which prevents the lipid membrane permeability and reduces the discharge of intracellular components, such as RNA and DNA, resulting in inhibiting the destruction of pathogens. Though, the electrostatic repulsion presents between the negatively charged surfactant molecules at low concentration and thus preserves the lipid membrane structures of pathogens [17]. It has been observed that the surfactant is biocompatible after using at a low concentration of 0.05 wt.%. This finding agrees based on the earlier studies conducted by [11,12] reporting that cationic surfactants are more toxic antibacterial agents at pH > 7, whereas anionic surfactants demonstrate antibacterial activity at pH < 7. In general, non-ionic nature of the surfactants do not display antibacterial activity [21]. As a result, it can be analyzed that a lower concentration of surfactant (SDBS) solution is appropriate for the dispersion of CNTs and additional toxicity studies of these nanotubes.

### 3.2. Antimicrobial Activity of CNTs

The antimicrobial activity of CNTs (DWCNTs and MWCNTs) was evaluated by assessing the growth curve of treated pathogens at OD600 nm. The OD growth curves of treated pathogens compared with different concentrations of modified DW and modified MW nanotubes in Figure 6 and Figure 7. In general, CNTs in bundles form do not produce any damage to the pathogens [20,34]. This non-antimicrobial activity of CNTs can be attributed to the larger diameter of tubes and poor solubility in the suspension as compared to modified CNTs [35]. It indicates that the proper dispersion of carbon nanotubes plays a vital role in their interaction with pathogens, as seen in (Figure 2). Because of the substantial vulnerability of surfactant (SDBS) to disperse carbon nanotubes, it was taken for further determination of the antimicrobial activity of CNTs.

Besides, SDBS was used as a biocompatible surfactant to check the microbial interactions with the nanotubes. The selected pathogens, such as *Staphylococcus aureus*, *Pseudomonas aeruginosa*, *Klebsiella pneumoniae,* and fungal strain *Candida albicans*, were incubated with the desired concentration of dispersed DWCNTs and MWCNTs in 0.05 wt.% surfactant. The impact of carbon nanotubes on microbial cell growth was studied. The OD growth curves at OD600 nm for these isolates have been shown in Figure 6 and Figure 7. At concentrations of 20, 40, and 60 μg/mL, there was a significant increase in all microbial cell growth following treatment with DW and MW carbon nanotubes for 24 h and 48 h.

In contrast, carbon nanotubes with a concentration of 20 μg/mL were showed a maximum value for microbial cell growth. Though nanotubes were significantly inhibited the microbial cell growth with a concentration of 80 and 100 μg/mL. *Pseudomonas aeruginosa* and *Klebsiella pneumoniae* were showed more susceptible to the carbon nanotubes with higher concentration. The results indicated that both DW and MW carbon nanotubes were showed their antimicrobial activity after functionalized with surfactant solution. As far as we know, there are limited published reports on the antimicrobial activity of DWCNTs. The previous studies [22,36,37] tell us about the antimicrobial activity of single-walled and multi-walled CNTs on different pathogenic strains, such as *E. coli, Enterococcus faecium,* and *Salmonella enteric*, but still lack information on the antimicrobial activity of DWCNTs. This study showed that DWCNTs possess the capacity to inhibit microbial cell growth and cause cell membrane damage. However, it can be seen that the antimicrobial activity of MWCNTs is higher against the pathogens as compared to DWCNTs. It may be due to the multiple layers of the graphene structure of MWCNTs [38]. Thus, we observed that both types of functionalized CNTs contain broad-spectrum antimicrobial effects.

### 3.3. Microbial Viability Based on Concentration and Treatment Time

The reduction in the number of viable cells after being treated with varying concentrations of SDBS-modified DWCNTs and MWCNTs, as seen in (Figure 8A,B). The antimicrobial property of surfactant-modified CNTs depends on the concentration [39]. After 24 h treatment with 20, 40, 60, 80, and 100 μg/mL MWCNTs, the viability of cells was decreased by 35, 49, 64, 75 and 83 percent, respectively, against *Staphylococcus aureus*. By contrast, the number of viable cells was decreased after applied the same concentration of DWCNTs against *Staphylococcus aureus*. As the concentration of both types of CNTs has increased, the reduction occurred in the viable cell number. Similarly, the viability of the pathogens, such as *Pseudomonas aeruginosa* was decreased 45, 59, 71, 86, and 95 percent by increasing concentration of MWCNTs, but the reduction has seen less in viable cells (*Pseudomonas aeruginosa*) after applied DWCNTs. These findings verify the previous studies that MWCNTs exhibits higher antimicrobial activity than DWCNTs [23,40].

By contract, SDBS-MWCNTs and SDBS-DWCNTs treatment were attained 39, 53, 69, 81, and 89 percent and 25, 35, 45, 66, and 73 percent reduction in viable cells number corresponding to the concentration of 20, 40, 60, 80, and 100 μg/mL, respectively, against *Klebsiella pneumoniae*. However, the viability of fungal strain, such as *candida albican* has been decreased 30, 44, 60, 72, and 80 percent and 15, 27, 35, 58, and 63 percent after 48 h of treatment with f-MWCNTs and f-DWCNTs. The number of viable cells was decreased after treating samples with surfactant-modified DWCNTs and MWCNTs, which showed the antimicrobial activity of these SDBS-modified CNTs. The reduction in viable cells number also reflects observations of delayed exponential log phases of these samples treated with surfactant-modified CNTs, confirming that f-MWCNTs exhibits more antimicrobial activity than f-DWCNTs against pathogens. Furthermore, it has also been studied the effect of treatment time on the reduction of microbial growth treated with f-CNTs. The decrease in the number of microbial growth after being treated with surfactant-modified CNTs at 100 μg/mL concentration for different treatment time has been shown in (Figure 8C,D). Both types of f-CNTs demonstrate a similar reduction in the number of microbial growth with respect to treatment time, where a reduction in the number of viable cells indicates a positive relationship with the treatment time. It was observed that a large number of viable cells decreased as the treatment time increased. Also, the effect of treatment time on microbial cell numbers was more obvious for f-MWCNTs. This indicates that the antimicrobial activity of functionalized MWCNTs contains a strong treatment time dependence.

Figure 9 indicates the effect of control samples on the viability of pathogens examined by the colony counting method, after being treated with different agents at a concentration of 100 μg/mL for 24 h and 48 h. It has been observed that both purified-CNTs and only the surfactant showed less inhibition of microbial growth after overnight incubation, suggested that unmodified-CNTs did not show any significant antimicrobial activity because nanotubes in bundle forms produce less antimicrobial effect to the pathogens [41]. It has also been examined that changes occur in the number of viable cells after pathogens treated under the same condition used to check the optical density (OD) growth curves. The result is shown in Figure 9, where no significant decreases in the number of viable cells were observed after treated with unmodified CNTs. The less antimicrobial efficiency of both types of CNTs can be ascribed to their functional and structural properties. It is obvious that unmodified CNTs are very hydrophobic in nature and hard to make its dispersion in aqueous solution because of van der Waals forces [30].

Based on the earlier reports [42,43], CNT’s antimicrobial mechanisms to the microbial cells were due to the cell membrane damage through direct interaction with CNTs. Thus, the dispersion of CNTs plays a very crucial role in the inactivation of microbial cells. Higher dispersion means CNTs strongly interact with the cells and thus, the cell death rate is significantly higher [13]. In fact, a good dispersion is very important premise for the CNTs to show higher microbial cells’ inactivation [41]. Hence, it is rational to attain the result that unmodified-CNTs did not demonstrate the antimicrobial activity to the pathogens. The surface modification of CNTs by non-covalent modification via surfactant molecules not only helps to improve the dispersion of CNTs, but also increases their antimicrobial activity [12]. The antimicrobial effect of surfactant solution (SDBS) has also been shown in (Figure 9). The surfactant (SDBS) was also exhibited the inactivation of the selected pathogens after overnight incubation. The rate of inhibition was observed 25, 35, 30, 21 percent, respectively.

In fact, modified-CNTs showed effective microbial cell inactivation compared to unmodified-CNTs. Interestingly, it has been shown that unmodified-MWCNTs showed higher antimicrobial activity than unmodified-DWCNTs. Also, it was obvious that the antimicrobial effectiveness of non-covalently modified CNTs strongly based on the applied molecules of surfactant (SDBS). The stronger antimicrobial efficiency of the surfactant molecules, the greater inactivation proficiency of the surfactant modified CNTs. This work provides a single platform to discuss the biocompatibility of surfactant and antimicrobial mechanisms of functionalized CNTs against different selected pathogens.

Figure 10 shows TEM images of purified-CNTs and SDBS-modified CNTs. The purified form of carbon nanotubes are closely packed with each other individually (a, c), whereas CNTs, after treated with SDBS become significantly untied, and highly dispersed in the surfactant solution without causing any CNT’s structural damage (b, d). This demonstrates that surfactant (SDBS) plays an important role in increasing the CNT’s dispersion power.

Figure 11 reveals that FE-SEM images of microbial cells interact with surfactant-modified CNTs (DWCNTs and MWCNTs). To explore the antimicrobial mechanisms of CNTs, FESEM was applied to assess the morphological changes and image the microbial surface after treatment. It was observed that control group microorganisms were intact in saline solution, and after 4 h of incubation maintained their outer membrane structural integrity (Figure 11). While after 4 h of treatment with 100 μg/mL of CNTs (DWCNTs and MWCNTs), the images showed extensive interaction between CNTs and microbial cell walls, causing damage to the outer membrane. The most obvious effects of CNTs were observed in *S. aureus*; the surface structure of bacterium changed from smooth to corrugated and even entirely disappeared (Figure 11).

The long CNTs, such as MWCNTs (Figure 11C) were observed wrapped around the surface of *S. aureus* (A), which led to more severe cell wall damage than DWCNTs (Figure 11B). In general, CNTs with long lengths provide more opportunity to wrap around the surface of pathogens and causing cell wall damage [35]. The needle-like actions were also observed in bacteria treated with DWCNTs (Figure 11B). Interestingly, with respect to Gram-negative bacteria, such as *Pseudomonas aeruginosa* and *Klebsiella pneumoniae* (Figure 11D,G), it was observed that both CNTs wrapped around the surface of bacteria and caused cell wall and membrane lysis (Figure 11E,F,H,I). According to our understanding, no reports published on the antimicrobial activity of DWCNTs towards *Klebsiella pneumoniae*. We found that CNTs contain broad-spectrum antimicrobial activity to all different selected pathogens. Previously, it has been reported that carbon nanotubes stick to the surface of microbial cells because of the electrostatic interactions [23]. The authors in [44] reported that microbial cells are more easily attached to the nanotubes’ structures, but the relationship between cellular adhesion and surface roughness of nanostructures still remains unclear.

It has been investigated that CNTs contain different antimicrobial activity against Gram-negative bacteria compared to Gram-positive bacteria due to different surface morphology of microbes. The authors in [45] have been used the atomic force microscopy to investigate the mechanical properties of microbial cell surface in aqueous solution. They reported that Gram-negative bacteria had a harder surface as compared to Gram-positive ones. Thus, the mechanical properties and different structural characteristics of the microbial cell wall could affect the antimicrobial activity of CNTs.

Finally, SEM was carried out for the surface analysis of yeast cells (*Candida albican*) treated with CNTs. Microscopic results showed that both types of CNT interact with the yeast cell (J) and form a web-like structure, which leads to cause cell wall damage by punctuations entering inside treated cells (Figure 11K,L). Thus, the data described here certainly argue that cell wall or membrane damage was an early consequence, which in turn induced a reduction in colony-forming units (CFU). It is worth mentioning that a different type of interaction occurs between CNTs and microbes, particularly depends on not only surface functional group or/and length of carbon nanotubes, but also morphological structure of microbial cells [23]. The authors in reference [42] reported that intrinsic antimicrobial mechanisms linked with CNTs’ length-dependent wrapping and diameter-dependent piercing on the microbial cell lysis.

However, it can be clearly observed at the edge that individually dispersed CNTs are attached at one end of microbial cells, and strictly adhered to the other end, which acts as needles surrounding the cells. In general, there are two reasons to increase the antimicrobial activity of carbon nanotubes. Firstly, the presence of molecules of surfactant, which facilitates the increase of the surface area for microbial interactions and favors the debundling of CNTs. Second, when a molecule of surfactant interacts with the cells, it ruptures and penetrates the cell membrane and ultimately causes cell death [42]. Earlier studies showed that CNTs after non-covalent modification had strong potential to adhere to the cell membrane of pathogens due to strong van der Waals interactions between individual nanotubes [35]. With respect to SWCNTs, nanotubes can capture the microbial cells and cause cell death because of direct interaction and physically punctures the outer membrane of the cells as a result [46,47,48].

Contrary to SWCNT, both DWCNTs and MWCNTs can also capture the microbial cells but do not effectively kill the pathogens, which is probably because of the larger diameter of DWCNTs and MWCNTs as compared to SWCNTs [49]. Akasaka and his colleague confirmed these findings that MWCNTs with a diameter ~30 nm had a strong potential to stick or adhere to the outer membrane of pathogens by physical sorption, which was not linked to the antimicrobial resistance [40]. Yang and his colleague were also reported that covalently-modified MWCNTs with –COOH and OH groups attached could form aggregates of cells without exhibited antimicrobial activity [22].

Concerning application potential, the benefits of killing or capturing pathogens by non-covalent-modified CNTs apparently include three considerations as compared to other antimicrobial agents. Firstly, it is identified that the mesopore volume of pristine CNTs and BET (Brunauer–Emmett–Teller) surface area lies in between the area of 250 m^2^/g and 0.85 cm^3^/g [50]. Thus, CNTs can provide large surface areas, which are able to immobilize the large biotic contaminants, such as bacteria and viruses. Secondly, biological contaminants and surfactant-modified CNTs are able to form aggregates and gradually deposit at the bottom [21]. This does not only purify the water phase rather also reduces the residual of surfactant-modified CNTs as antimicrobial agents, preventing second contamination. Thirdly, the surfactant-modified CNTs offer simultaneous inactivation and capture of biological contaminants, but other carbon-based nanomaterials can also offer capture of biological contaminants [21].

## 4. Conclusions

In this study, the dispersion potential of surfactant-modified CNTs and their antimicrobial activity to *Staphylococcus aureus*, *Pseudomonas aeruginosa*, *Klebsiella pneumoniae,* and fungal strain *Candida albicans* were investigated. Both types of CNT—DWCNTs and MWCNTs—can inhibit the growth of tested pathogens. Particularly, CNTs may selectively damage the walls or membranes of microbes, depending on not only the functional groups and length of nanotubes but also on the shapes of pathogens. Long CNTs may wrap around the surface of pathogens and increase the surface contact area with the cell wall of microbes. The dispersion of CNTs was observed by UV–vis absorption, FTIR, and TEM images indicate that SDBS-treated CNTs have the ability to disperse in the aqueous phase. However, optical density growth curve and the number of viable cells confirmed that surfactant-modified CNTs showed high antimicrobial activity. Besides, FESEM images indicated the strong type of interactions present between SDBS-treated CNTs and microbial cells. The molecules of the surfactant bind on the surface of graphene sheets, which helps to disperse nanotubes in aqueous solutions. The stronger dispersion of CNTs increased its antimicrobial activity. MWCNTs contained higher antimicrobial activity as compared to DWCNTs.

## Figures and Tables

**Figure 1 materials-13-01676-f001:**
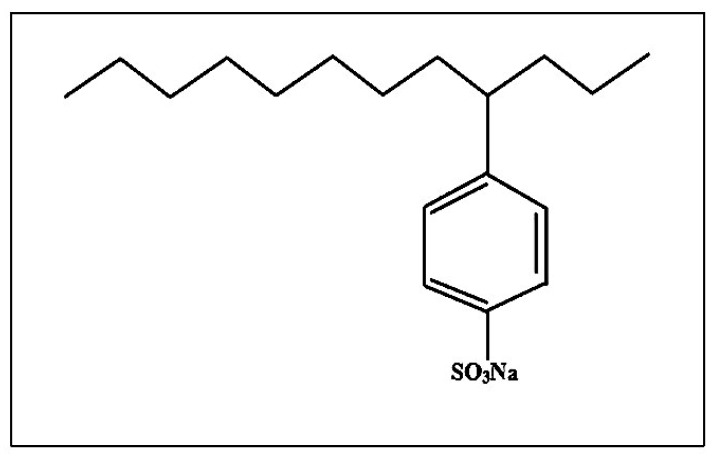
The chemical structure of surfactant-sodium dodecylbenzene sulfonate (SDBS) [8].

**Figure 2 materials-13-01676-f002:**
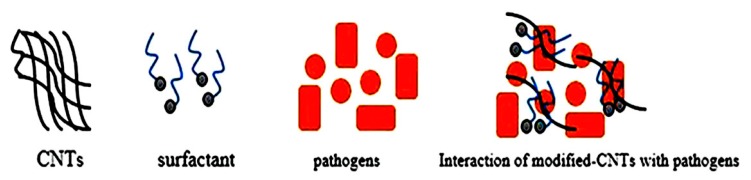
Schematic illustration of surfactant-modified carbon nanotubes (CNTs) and their possible interaction with the pathogens.

**Figure 3 materials-13-01676-f003:**
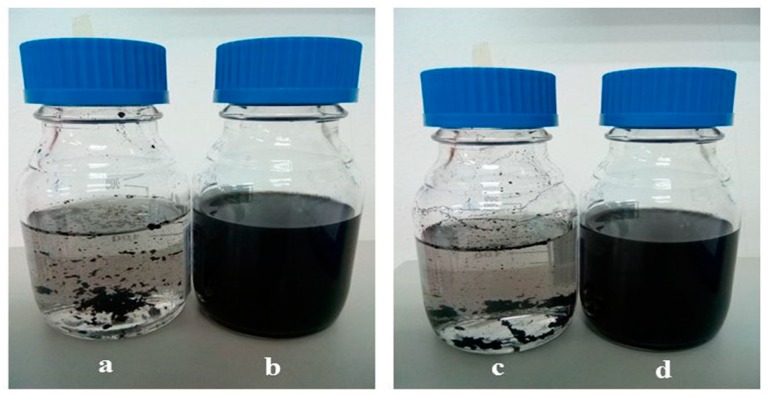
Carbon nanotubes suspension (**a**) sonication of multi-walled carbon nanotubes (MWCNTs) in distilled water (DW), (**b**) sonication of MWCNTs in surfactant (SDBS), (**c**) sonication of double-walled carbon nanotubes (DWCNTs) in DW, and (**d**) sonication of DWCNTs in surfactant (SDBS) for 30 min.

**Figure 4 materials-13-01676-f004:**
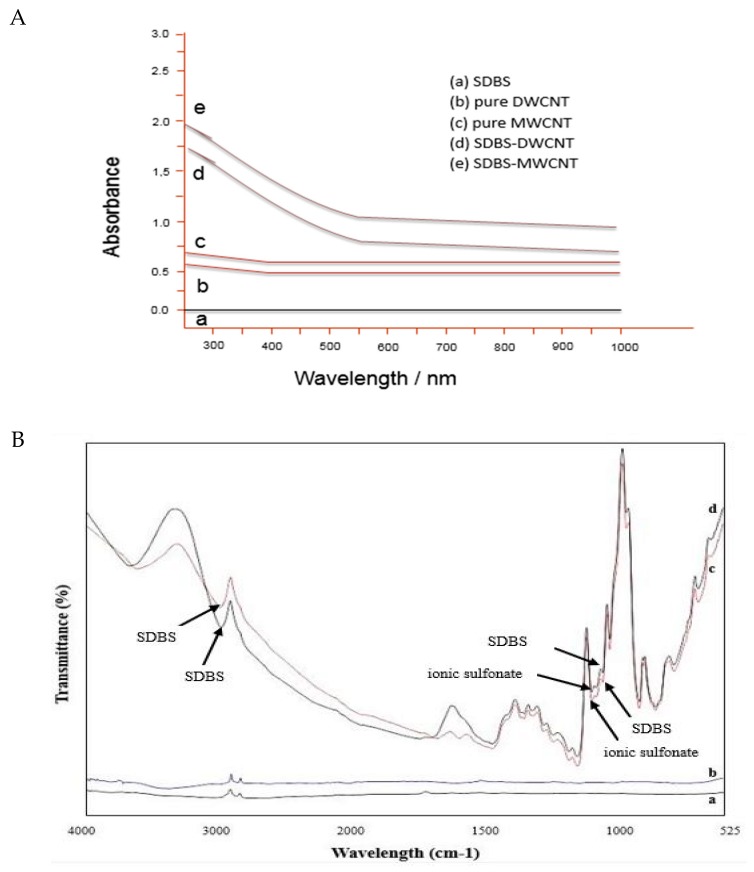
(**A**) Uv-vis spectra of carbon nanotubes suspension; (**B**) Fourier transform infrared spectroscopy (FTIR) spectra of purified-DWCNT (a), purified-MWCNT (b), DWCNT-SDBS (c), and MWCNT-SDBS (d).

**Figure 5 materials-13-01676-f005:**
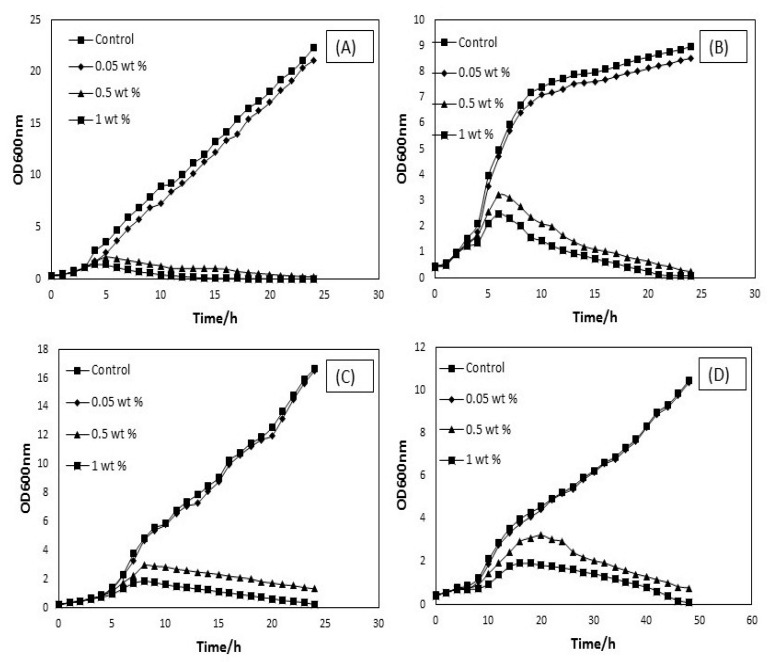
Optical density (OD) growth curves of incubated *Staphylococcus aureus* (**A**), *Pseudomonas aeruginosa* (**B**), *Klebsiella pneumonia* (**C**), and *Candida albican* (**D**) with different concentration of SDBS.

**Figure 6 materials-13-01676-f006:**
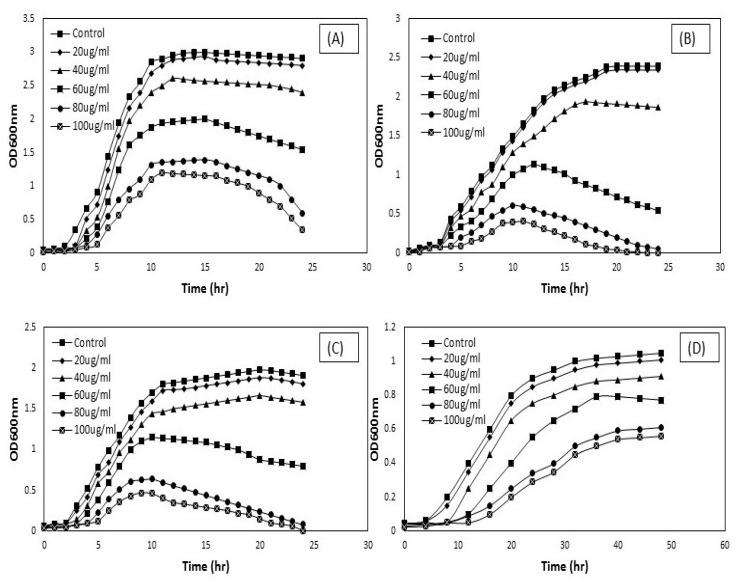
OD growth curves attained when 150 μL of ~10^7^–10^8^ CFU/mL *Staphylococcus aureus* (**A**), *Pseudomonas aeruginosa* (**B**), *Klebsiella pneumoniae* (**C**), and *Candida albican* (**D**), treated with MWCNTs at the following conditions and then grown in 5 mL of LB and YPD broth at 37 °C: treated with desired concentration of 20, 40, 60, 80, and 100 μg/mL MWCNTs in DI water for 1 h. The cell suspension was used as a control sample in DI water.

**Figure 7 materials-13-01676-f007:**
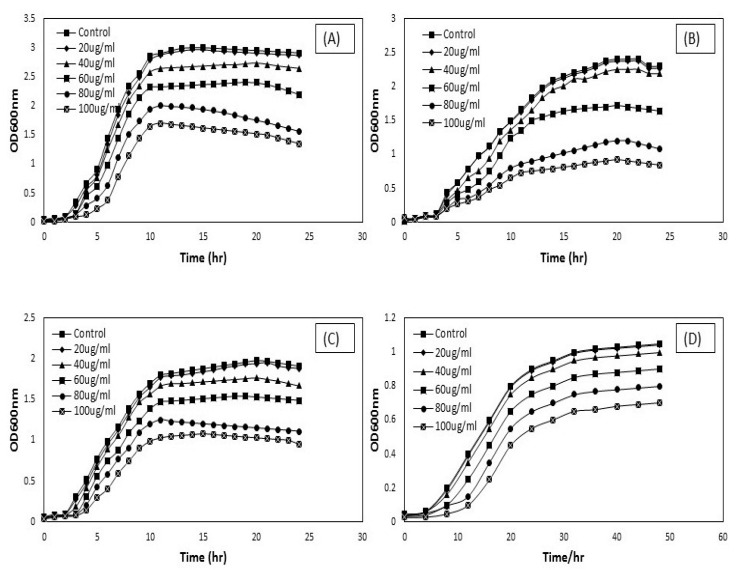
OD growth curves attained when 150 μL of ~10^7^–10^8^ CFU/mL *Staphylococcus aureus* (**A**), *Pseudomonas aeruginosa* (**B**), *Klebsiella pneumoniae* (**C**), and *Candida albican* (**D**) was treated with DWCNTs at the following conditions and then cultivated in 5 mL of LB and YPD broth at 37 °C: treated with desired concentration of 20, 40, 60, 80, and 100 μg/mL DWCNTs in DI water for 1 h. The cell suspension was used as a control sample in DI water.

**Figure 8 materials-13-01676-f008:**
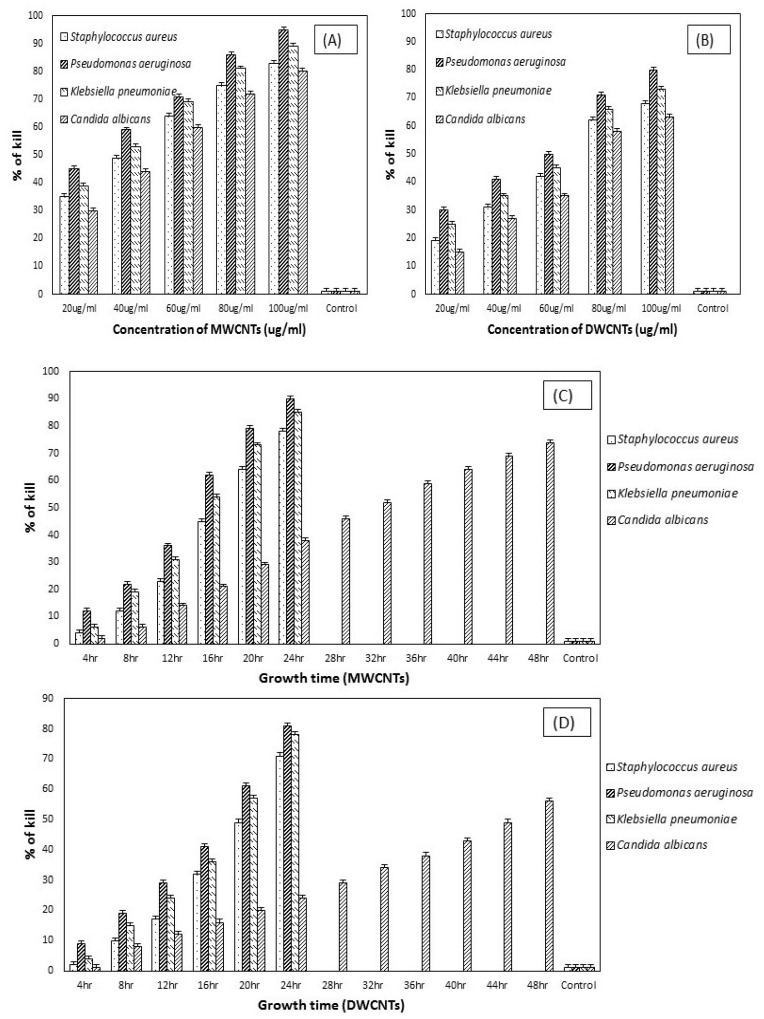
Antimicrobial activity of MWCNTs and DWCNTs on microbial cells based on concentration and treatment time. Microbial viability assay was carried out after incubation of microbial cells (~10^7^–10^8^ CFU/mL) with desired concentration (**A**,**B**) and treatment time at 100 μg/mL (**C**,**D**) of MWCNTs and DWCNTs for 24 h in case of *Staphylococcus aureus*, *Pseudomonas aeruginosa*, and *Klebsiella pneumoniae* and 48 h for *Candida albicans*. The survival of cells was examined by a colony counting method and stated as a percentage with respect to microbial cells (untreated) incubated with DW. The control was microbial cells treated with distilled water.

**Figure 9 materials-13-01676-f009:**
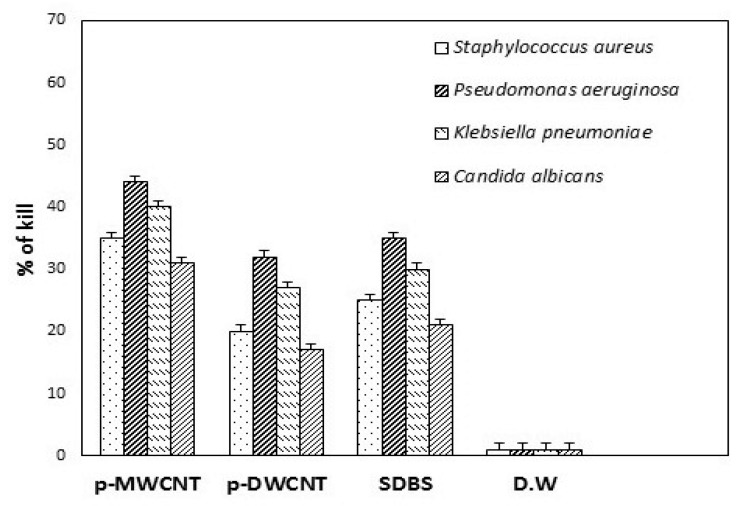
Effect of control samples on the viability of pathogens examined by the colony counting method, after being treated with purified-MWCNT, purified-DWCNTs, and SDBS at a concentration of 100 μg/mL or DW only (UT, untreated) for 24 h for *Staphylococcus aureus*, *Pseudomonas aeruginosa*, and *Klebsiella pneumoniae* and 48 h for *Candida albicans*.

**Figure 10 materials-13-01676-f010:**
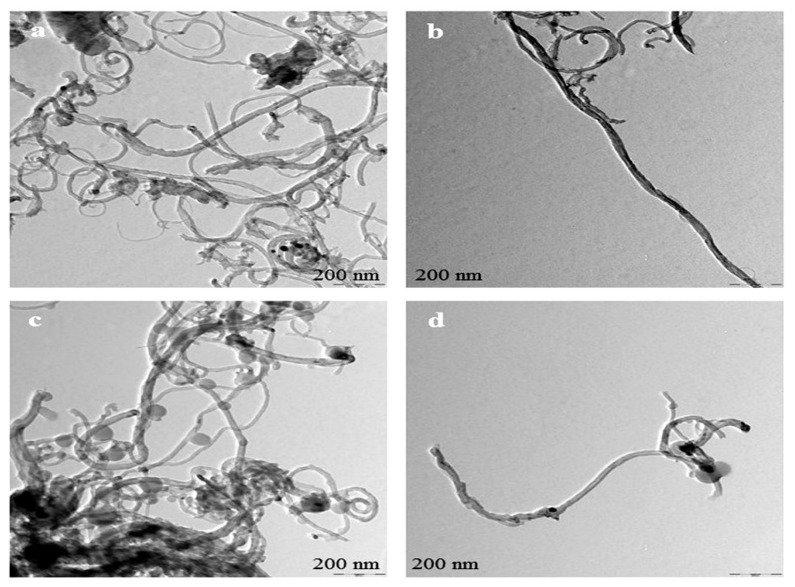
Transmission electron microscopy (TEM) images of (**a**) pure DWCNTs, (**b**) SDBS-treated DWCNTs, (**c**) pure MWCNTs, and (**d**) SDBS-treated MWCNTs.

**Figure 11 materials-13-01676-f011:**
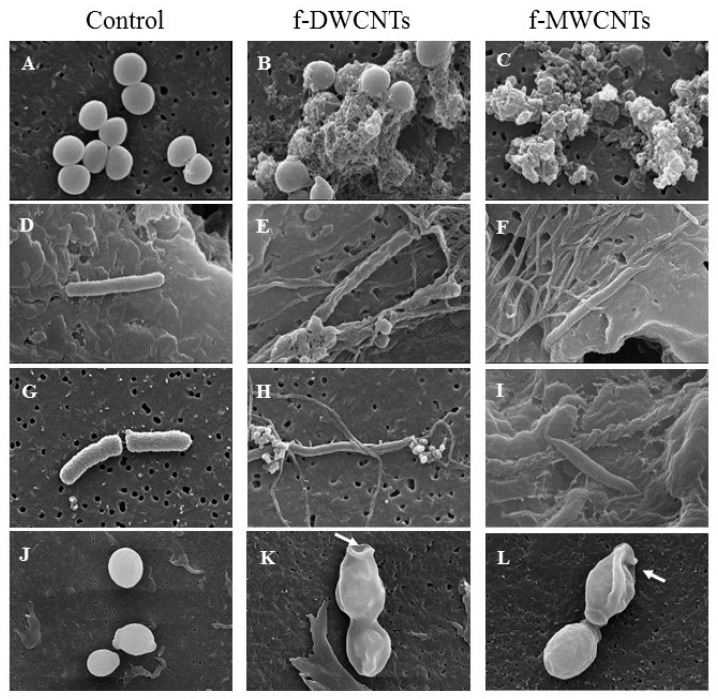
SEM images of (**A**–**C**) *Staphylococcus aureus*, (**D**–**F**) *Pseudomonas aeruginosa*, (**G**–**I**) *Klebsiella pneumoniae*, and (**J**–**L**) *Candida albican* microbial cells. The images refer to the untreated control group and microbial cells exposed to 100 μg/mL f-DWCNTs and f-MWCNTs at 80,000 × magnification. The arrows indicate the CNTs’ web.

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
