# Peer review of "Elucidation of Antimicrobial Activity of Non-Covalently Dispersed Carbon Nanotubes"

_materials, 2020, doi:10.3390/ma13071676_

Round 1

Reviewer 1 Report

In this ms., the authors described results of their studies on antibacterial in vitro potential of DWCNTs and MWCNTs dispersed in solution due to the presence of surfactant SDBS, molecules of which were adsorbed onto CNTs surface. This is a solid work, the experiments were designed and performed properly and their results were appropriately interpreted. Unfortunately, in my opinion the main problem with this ms. is its novelty. A few previous works, including those cited in this ms. (Ref. 11-13) already showed that different CNTs dispersed in solution due to surface functionalization with different surfactants (also SDBS), exhibit antibacterial activity. Therefore, the authors should make more clear what is really new in their study. The only aspect I am able to note is a broader antibacterial spectrum than those studied in previous works.

Appropriate explanations should be introduced in the Results & Discussion section.

Some references in the body tet are not appropriate:

line 74, Ref 15 and 16

line 84, Ref 16

line 261, no reference numbers

Moreover, the References section should be carefully corrected.

References are not uniform. In some of them the journal title is provided in full, in some other ones - abbreviated.

In some references "pp." abbreviation is present but in some other ones is missing.

Refs. 4 and 6 - page numbers missing

Author Response

Sr.no

Reviewer 1 comments

Response

1

In this ms., the authors described results of their studies on antibacterial in vitro potential of DWCNTs and MWCNTs dispersed in solution due to the presence of surfactant SDBS, molecules of which were adsorbed onto CNTs surface. This is a solid work, the experiments were designed and performed properly and their results were appropriately interpreted. Unfortunately, in my opinion the main problem with this ms. is its novelty. A few previous works, including those cited in this ms. (Ref. 11-13) already showed that different CNTs dispersed in solution due to surface functionalization with different surfactants (also SDBS), exhibit antibacterial activity. Therefore, the authors should make more clear what is really new in their study. The only aspect I am able to note is a broader antibacterial spectrum than those studied in previous works.

We thank the reviewers for the valuable comments. All the major revisions as suggested by the reviewers have been addressed accordingly. As per the clarification on novelty, to the best of our knowledge, the antimicrobial activity of DWCNTs to Klebsiella pneumoniae has not been investigated. We have been observed that CNTs, DWCNTs and MWCNTs both have broad-spectrum antimicrobial effects. Also, this study has provided such a single platform to discuss the biocompatibility of surfactant (SDBS) and the antimicrobial activity of functionalized-CNTs (DWCNTs and MWCNTs) to different selected pathogens. The novelty has been addressed in the Introduction section Line No. 80-87 and Results and Discussion section Line No. 303-311.

2

Appropriate explanations should be introduced in the Results & Discussion section.

The appropriate explanation has also given under the section Results and Discussion. For instance, As far as we know, there are limited published reports on the antimicrobial activity of DWCNTs. The previous studies [18, 49, 50] tell us about the antimicrobial activity of single-walled and multi-walled CNTs on different pathogenic strains, such as E. coli, Enterococcus faecium, and Salmonella enteric, but still lacks to know about the antimicrobial activity of DWCNTs. This study has been shown that DWCNTs possess the capacity to inhibit microbial cell growth and cause cell membrane damage. But, it can be seen that the antimicrobial activity of MWCNTs is higher against the pathogens as compared to DWCNTs. It may be due to the multiple layers of the graphene structure of MWCNTs [30]. Thus, we observed that both types of functionalized-CNTs contain broad-spectrum antimicrobial effects in Line No. 303-311, Line No. 383-385, Line No. 426-428.

3

Some references in the body tet are not appropriate:

line 74, Ref 15 and 16

line 84, Ref 16

line 261, no reference numbers

Introduction Section

In the body text references, we have given a proper explanation for the reference 15 and 16, for instance, the molecules of surfactants are strongly adsorbed on the CNTs surface and inhibit their re-agglomeration so that dispersions of nanotubes could maintain colloidal stability for several months [15, 16] in Line No. 72-74.

We have replaced the Reference 16 to References 7 and 43 in Line No. 90.

Results and Discussion Section

We have included the reference number [11, 12] in Line No. 271.

4

Moreover, the References section should be carefully corrected.

References are not uniform. In some of them the journal title is provided in full, in some other ones - abbreviated.

In some references "pp." abbreviation is present but in some other ones is missing.

Refs. 4 and 6 - page numbers missing

References Section

We have carefully revised this section in a more comprehensive way as per suggestion. See the References section in Line No. 524-672.

We have removed the “pp.” abbreviation throughout the references section.

As per suggestion, we have included the page numbers of references 4 and 6 in Line No. 330-335.

Reviewer 2 Report

Summary

This paper has nothing new from the already published. However, it is noteworthy the effort of the authors in performing the antimicrobial test in an organized and following the ASTM standards. A lot of work was done, but unfortunately, for me, others have already done the same or similar to them.

Broad comments

This work as not novelty as published by the authors. There are more than 17000 works published in the internet. All the protocol and even the pathogens are not new. For example, Arias and Yang (2009)[1] investigated the antimicrobial activities of SWCNTs and MWCNTs with different surface groups towards rod-shaped or round-shaped gram-negative and gram-positive bacteria. In another work, Dong et al (2012)[2] investigated the antibacterial properties of SWCNTs dispersed in different surfactant solutions (sodium holate, sodium dodecyl benzenesulfonate, and sodium dodecyl sulfate) against Salmonella enteric (S. enteric), E. coli, and Enterococcus faecium. In addition the antimicrobial mechanism of action of CNTs on different pathogens was also described[3]. So, I would like to see in this manuscript something different from the already published.

The introduction section has changes to be made and the results section needs to be well supported using updated literature. The references are too old.

Additionally, the English spelling is not good and a lot of mistakes and faults were found in the text. I have made some appointments but many more must be done. Otherwise, it is difficult to read and understand the ideas. Also, it is important also to use always the correct format for units such as liter (L) or milliliters (mL), weight percent (wt%), micrograms (µg) and so one.

Specific comments

Introduction

Line 44-45 must be changed to: "Carbon nanotubes first discovered in 1991 by a Japanese scientist Sumio Lijima, are currently considered…"

Line 46 is not rational since mechanical, electronic, structural properties are examples of physicochemical properties. The author must correct the sentence.

Line 93 is not correct. "It must be: The method used for the purification of CNTs is as follows."

Line 95 is not correct. "After heating, CNTs were presented inserted into a conical flask comprising 6M HCl for the 95 eradication of metallic catalyst (Ni, Fe, etc.)."

Line 96-98 must be changed. "Later, a membrane filter was used for the filtration of the acidic solution, and already the filtered nanomaterials were shifted into a conical flask containing 3M NaOH and further heated at temperature (100°C) under reflux to eliminate the aluminum oxides."

Line 104 should be corrected. "Ten milligrams 10 mg of CNTs after purification were supplemented suspended with 0.05 wt.% weight concentration of sodium dodecylbenzene sulfonate (SDBS, purchased from Sigma–Aldrich) solution."

Line 107 must be changed. "The resulting suspension was centrifuged for one hour at 10,000rpm."

Line 109-111 The text is not correctly spelling. It must be changed.

Experimental Section

Some notes were done in the manuscript. Others are highlighted. These mean that some changes should be done.

Results and Discussion

Line 223 The authors must change to "It has been shown that solubility…"

Line 244-246 It is missing a reference to support the sentence "It has been shown that the antimicrobial property of surfactant (SDBS) depends on concentration and treatment time."

All the figures have low resolution and it is difficult to discriminate the different samples results.

The discussion of the results is well done. Even though some references to support their work is missing.

Conclusions

No comments.

References

Authors should update the literature, as there are many references before 2015. For the last five years, they mentioned only 5 references out of 43, which is very poor concerning the high activity in this research field.

The reviewer,

[1] Arias, L.R., Yang, L. (2009), Inactivation of bacterial pathogens by carbon nanotubes in suspensions. Langmuir, 25(5):3003–3012.

[2] Dong, L., Henderson, A., Field, C. (2012), Antimicrobial activity of single-walled carbon nanotubes suspended in different surfactants. Journal of Nanotechnology, 2012:1–7.

[3] Dizaj, S.M., Mennati, A., Jafari, S., Khezri, K., Adibkia, K. (2015), Antimicrobial Activity of Carbon-Based Nanoparticles, Advanced Pharmaceutical Bulletin, 5(1): 19-23.

Author Response

Sr.no

Reviewer 2 comments

Response

1

This paper has nothing new from the already published. However, it is noteworthy the effort of the authors in performing the antimicrobial test in an organized and following the ASTM standards. A lot of work was done, but unfortunately, for me, others have already done the same or similar to them.

As per the clarification on novelty, previous works were particularly focused on the types of CNTs, such as Single-walled and Multi-walled nanotubes. There are very limited reports on the antimicrobial activity of Double-walled CNTs. Previous literature does not provide such a single platform to discuss about the biocompatibility of surfactant (SDBS) and antimicrobial activity of CNTs against different selected pathogens. To the best of our knowledge, the antimicrobial activity of DWCNTs to Klebsiella pneumoniae has not been investigated. We have been observed that CNTs, DWCNTs and MWCNTs both have broad-spectrum antimicrobial effects. Also, this study has provided such a single platform to discuss the biocompatibility of surfactant (SDBS) and the antimicrobial activity of functionalized-CNTs (DWCNTs and MWCNTs) to different selected pathogens. The novelty has been addressed in the Introduction section Line No. 80-87 and Results and Discussion section Line No. 303-311.

2

This work as not novelty as published by the authors. There are more than 17000 works published in the internet. All the protocol and even the pathogens are not new. For example, Arias and Yang (2009)[1] investigated the antimicrobial activities of SWCNTs and MWCNTs with different surface groups towards rod-shaped or round-shaped gram-negative and gram-positive bacteria. In another work, Dong et al (2012)[2] investigated the antibacterial properties of SWCNTs dispersed in different surfactant solutions (sodium holate, sodium dodecyl benzenesulfonate, and sodium dodecyl sulfate) against Salmonella enteric (S. enteric), E. coli, and Enterococcus faecium. In addition the antimicrobial mechanism of action of CNTs on different pathogens was also described[3]. So, I would like to see in this manuscript something different from the already published.

According to our understanding, previous works were particularly focused on the types of CNTs, such as Single-walled and Multi-walled nanotubes, for instance, Arias and Yang (2009), Dong et al. (2012), and Dizaj et al. 2015. There are very limited reports on the antimicrobial activity of Double-walled CNTs. Previous literature does not provide such a single platform to discuss about the biocompatibility of surfactant (SDBS) and antimicrobial activity of CNTs against different selected pathogens. To the best of our knowledge, the antimicrobial activity of DWCNTs to Klebsiella pneumoniae has not been investigated. We have been observed that CNTs, DWCNTs and MWCNTs both have broad-spectrum antimicrobial effects. Also, this study has provided such a single platform to discuss the biocompatibility of surfactant (SDBS) and the antimicrobial activity of functionalized-CNTs (DWCNTs and MWCNTs) to different selected pathogens. The novelty has been addressed in the Introduction section Line No. 80-87 and Results and Discussion section Line No. 303-311, 383-385 and 426-428.

3

The introduction section has changes to be made and the results section needs to be well supported using updated literature. The references are too old.

We have made some changes in the Introduction Section, for instance, We have included some details as suggested in Line No. 44-47, 72-76, 78, 80-87. Also, we have included some Latest References, for instance, [44, 45] in Line No. 76, [46] in Line No. 89, [43] in Line No. 90.

Results and Discussion Section

As per the suggestion, we have included the Latest References in this section. For instance, Ref. [20-24] in Line No. 211-212, Ref. [47] in Line No. 238, Ref. [26-27] in Line No. 241, Ref. [28] in Line No. 243, Ref. [29] in Line No. 246, 247, Ref. [43] in Line No. 273, Ref. [29-46] in Line No. 284, Ref. [18, 49-50] in Line No. 304, Ref. [48] in Line No. 360, Ref. [47] in Line No. 366, Ref. [48] in Line No. 371.

4

Additionally, the English spelling is not good and a lot of mistakes and faults were found in the text. I have made some appointments but many more must be done. Otherwise, it is difficult to read and understand the ideas. 

As per the suggestions, the English spelling and grammer has been thoroughly checked and corrected throughout the manuscript.

5

Also, it is important also to use always the correct format for units such as liter (L) or milliliters (mL), weight percent (wt%), micrograms (µg) and so one.

We have revised the correct format for the units throughout the manuscript, as suggested by the reviewer, for instance, 10 mg in Line No. 112, 100 mL in Line No. 149, CFU/mL in Line No. 160, 150 μL in Line no. 160, (0.1 mL) in Line No. 183, 0.05 wt.%, 0.5 wt.%, and 1 wt.% in Line No. 252, 253, 256, 257, 261, 270, 20, 40, and 60 μg/mL in Line No. 295, 80 and 100 μg/mL in Line No. 300, CFU/mL in Line No. 313, 20, 40, 60, 80, and 100 μg/mL in Line No. 316, CFU/mL in Line No. 319, 20, 40, 60, 80, and 100 μg/mL in Line No. 322, 100 μg/mL in Line No. 348, 356, 390, 398, 416, 453.

6

Specific comments

Introduction

Line 44-45 must be changed to: "Carbon nanotubes first discovered in 1991 by a Japanese scientist Sumio Lijima, are currently considered…"

Introduction Section

The sentence has been changed as suggested. “Carbon nanotubes first discovered in 1991 by a Japanese scientist Sumio Iijima [1], are currently considered to be a top-class topic in academic research institutions and several industrial areas,” in Line No. 44, 45.

7

Line 46 is not rational since mechanical, electronic, structural properties are examples of physicochemical properties. The author must correct the sentence.

We have corrected the sentence as suggested. “Their impressive physicochemical properties are due to their incredible thermal and electrical conductivity, strong mechanical strength, and high aspect ratio of nanotubes [2-8]” in Line No. 45-47.

8

Line 93 is not correct. "It must be: The method used for the purification of CNTs is as follows."

We have revised the sentence as suggested. “The method used for the purification of CNTs is as follows,” in Line No. 101, 102.

9

Line 95 is not correct. "After heating, CNTs were presented inserted into a conical flask comprising 6M HCl for the 95 eradication of metallic catalyst (Ni, Fe, etc.)."

We have revised the sentence as suggested. “After heating, CNTs were inserted into a conical flask comprising 6M HCl for the eradication of metallic catalyst (Ni, Fe, etc.),” in Line No. 103, 104.

10

Line 96-98 must be changed. "Later, a membrane filter was used for the filtration of the acidic solution, and already the filtered nanomaterials were shifted into a conical flask containing 3M NaOH and further heated at temperature (100°C) under reflux to eliminate the aluminum oxides."

We have revised the sentence as suggested. “Later, a membrane filter was used for the filtration of the acidic solution, and the filtered nanomaterials were shifted into a conical flask containing 3M NaOH and further heated at temperature (100°C) under reflux to eliminate the aluminium oxides [11],” in Line No. 104-106.

11

Line 104 should be corrected. "Ten milligrams 10 mg of CNTs after purificationwere supplemented suspended with 0.05 wt.% weight concentration of sodium dodecylbenzene sulfonate (SDBS, purchased from Sigma–Aldrich) solution."

We have revised the sentence as suggested. “10 mg of CNTs after purification were suspended with 0.05 wt.% of sodium dodecylbenzene sulfonate (SDBS, purchased from Sigma–Aldrich) solution,” in Line No. 112, 113.

12

Line 107 must be changed. "The resulting suspension was centrifuged for one hour at 10,000rpm."

We have revised the sentence as suggested. “The resulting suspension was centrifuged for one hour at 10,000 rpm,” in Line No. 115, 116.

13

Line 109-111 The text is not correctly spelling. It must be changed.

We have revised the sentence as suggested. “Otherwise, a membrane filter (0.45 μm) was used to filter the solution and then distilled water used to wash the filtered nanomaterials until pH becomes neutral and dried the suspension in oven at 55°C [13],” in Line No. 117-119.

14

Experimental Section

Some notes were done in the manuscript. Others are highlighted. These mean that some changes should be done.

Experimental Section

As per the suggestion, we have made some changes to this section. For instance, we have critically checked the Grammer and Spelling mistakes throughout the manuscript and revised the format of units as explained above under the point 5.

15

Results and Discussion

Line 223 The authors must change to "It has been shown that solubility…"

Results and Discussion

We have revised the sentence as suggested. “It has been shown that solubility of both types of CNTs (DWCNT & MWCNT) was higher after applying a surfactant solution than that in their pure form,” in Line No. 232, 233.

16

Line 244-246 It is missing a reference to support the sentence "It has been shown that the antimicrobial property of surfactant (SDBS) depends on concentration and treatment time."

We have revised the sentence as suggested. “It has been shown that the antimicrobial property of surfactant (SDBS) depends on concentration and treatment time [11],” in Line No. 253-255.

17

All the figures have low resolution and it is difficult to discriminate the different samples results.

As suggested, all the figures have been improved with better resolution and quality. Now, we have improved the quality of images with better resolution.

18

The discussion of the results is well done. Even though some references to support their work is missing.

Results and Discussion Section

As per the suggestion, we have included the missing References in this section. For instance, Ref. [20-24] in Line No. 211-212, Ref. [47] in Line No. 238, Ref. [26-27] in Line No. 241, Ref. [28] in Line No. 243, Ref. [29] in Line No. 246, 247, Ref. [43] in Line No. 273, Ref. [29-46] in Line No. 284, Ref. [18, 49-50] in Line No. 304, Ref. [48] in Line No. 360, Ref. [47] in Line No. 366, Ref. [48] in Line No. 371, Ref. [36] in Line No. 423.

19

Conclusions

No comments.

Noted with thanks.

20

References

Authors should update the literature, as there are many references before 2015. For the last five years, they mentioned only 5 references out of 43, which is very poor concerning the high activity in this research field.

As per the suggestion, we have included the latest references as mentioned above under the point 3 and 18.

Reviewer 3 Report

In my opinion, it is a great paper, very well written, very complete, reporting on a very important and interesting topic. The research was well conducted, all results are well explained and discussed.

I just have some minor comments:

On Introduction, a few times, authors write "carbon nanotube" (singular), when it should be "carbon nanotubes" (plural) - e.g. lines 44 and 48.

Authors cite previous work done with CNTs for inhibition of pathogens growth (refs. 15-17, lines 73-77), however, the text is mainly focused on the problem of dispersion. Authors do not say what ref. 15-17 are all about, namely, what pathogens were studied there and what is the novelty of the present work, compared to the others, and that should be better stressed out and made clearer.

Why were double walled and multi walled CNTs studied and not single walled?

Minor points:

Figure 1 is too large, no need for such a huge image.

Caption of Fig4a is covering part of the text (line 211)

Caption of Fig4b should be in the same page as the figure.

Author Response

Sr.no

Reviewer 3 comments

Response

1

In my opinion, it is a great paper, very well written, very complete, reporting on a very important and interesting topic. The research was well conducted, all results are well explained and discussed.

Noted with thanks.   

2

I just have some minor comments:

On Introduction, a few times, authors write "carbon nanotube" (singular), when it should be "carbon nanotubes" (plural) - e.g. lines 44 and 48.

Introduction Section

As per useful suggestions by the reviewer, we have revised the “Carbon Nanotube” to “Carbon Nanotubesin Line No. 48, 59, 60, 61, 67.

3

Authors cite previous work done with CNTs for inhibition of pathogens growth (refs. 15-17, lines 73-77), however, the text is mainly focused on the problem of dispersion. Authors do not say what ref. 15-17 are all about, namely, what pathogens were studied there and what is the novelty of the present work, compared to the others, and that should be better stressed out and made clearer.

We have revised this part as suggested by the reviewer. For instance, “The molecules of surfactants are strongly adsorbed on the CNTs surface and inhibit their re-agglomeration so that dispersions of nanotubes could maintain colloidal stability for several months [15, 16]. Earlier studies have shown that CNTs are more effective in inhibiting the growth of pathogens (such as V. parahaemolyticus and Escherichia coli) in their dispersed form than CNT's bundles [44, 45]. The reason for the high effectiveness of individual nanotubes in inhibiting the microbial growth after dispersion could be because of the increased surface contact chances with the microbial cells (E. coli, S. typhimurium and P. denitrificants) [17]. The proposed antimicrobial activity of CNTs depends on their state of dispersion,” in Line No. 72-79.

As per the clarification on novelty, to the best of our knowledge, the antimicrobial activity of DWCNTs to Klebsiella pneumoniae has not been investigated. We have been observed that CNTs, DWCNTs and MWCNTs both have broad-spectrum antimicrobial effects. Also, this study has provided such a single platform to discuss the biocompatibility of surfactant (SDBS) and the antimicrobial activity of functionalized-CNTs (DWCNTs and MWCNTs) to different selected pathogens. The novelty has been addressed in the Introduction section Line No. 80-87 and Results and Discussion section Line No. 303-311.

4

Why were double walled and multi walled CNTs studied and not single walled?

The reasons for the selection of these two types of CNTs (DWCNTs and MWCNTs). These CNTs are easily available and cost-effective. There are limited studies on the antimicrobial activity of DWCNTs.

5

Minor points:

Figure 1 is too large, no need for such a huge image.

As advised, we have revised the Figure 1 and resized this image. Line No. 122-124.

6

Caption of Fig4a is covering part of the text (line 211).

We have adjusted the Caption of Fig 4a as suggested in Line No. 218.

7

Caption of Fig4b should be in the same page as the figure.

We have adjusted the Caption of Fig 4b as suggested in Line No. 231.

Round 2

Reviewer 2 Report

I do not have many comments to add to my previous revision. The authors have made an effort to increase the quality of the manuscript based on the referees' comments. However, in the references section is still missing the mention of the recent publications (only 35 % of the mentioned papers are within the last 5 years, which is very low). In addition, I have difficult in seeing the novelty and the utility of the manuscript to the audience. But, the decision to publish or not should be done by the editor.